# RETROSYNTHESIS PREDICTION VIA SEARCH IN (HYPER) GRAPH

## ABSTRACT

Retrosynthesis prediction is a fundamental challenge in organic synthesis, involving the prediction of reactants based on a given core product. Recently, semi-template-based methods and graph-edits-based methods have achieved good performance in interpretability and accuracy. However, their mechanisms still fail to predict complex reactions, e.g., reactions with multiple reaction center or attaching the same leaving group to more than one atom. Hence, we propose, a semi-template-based method, the **Retro**synthesis via **S**earch **i**n (Hyper) **G**raph (RetroSiG) framework to alleviate these limitations. In this paper, we cast the reaction center identification and the leaving group completion as search in the product molecular graph and leaving group hypergraph respectively. RetroSiG has several advantages as a semi-template-based method: First, RetroSiG is able to handle the complex reactions mentioned above with its novel search mechanism. Second, RetroSiG naturally exploits the hypergraph to model the implicit dependencies between leaving groups. Third, RetroSiG makes full use of the prior, i.e., one-hop constraint. It reduces the search space and enhances overall performance. Comprehensive experiments demonstrate that RetroSiG achieves a competitive result. Furthermore, we conduct experiments to show the capability of RetroSiG in predicting complex reactions. Ablation experiments verify the effectiveness of individual components, including the one-hop constraint and the leaving group hypergraph.

## 1 INTRODUCTION

The retrosynthesis prediction is to identify reactants that can be used to synthesize a specified product molecule. It is first formalized by Corey (1991) and now becomes one of the fundamental problems in organic chemistry. However, the problem is challenging as the search space of all possible transformations is huge by nature Gothard et al. (2012). Hence, people have been seeking various computer algorithms to assist experienced chemists. Among them, machine learning based methods have played a vital role and achieved significant progress in this area recently Coley et al. (2017).

The template-based and template-free methods have their own set of advantages and disadvantages. The template-based methodologies Coley et al. (2017); Dai et al. (2019); Segler & Waller (2017); Chen & Jung (2021); Yan et al. (2022) tackle retrosynthesis prediction as a template retrieval problem. Following template retrieval, these approaches utilize cheminformatics tools like RDKit Landrum (2006) to construct full reactions. Despite their high interpretability and assurance of molecule validity, they cannot predict reactions outside the template library. In contrast, the template-free approaches leverage deep generative models to produce reactants based on a given product directly. Since molecules can be represented using both graphs and SMILES sequences, existing techniques reframe retrosynthesis as either a sequence-to-sequence problem Lin et al. (2020); Zheng et al. (2019); Tetko et al. (2020); Seo et al. (2021); Wan et al. (2022) or a graph-to-sequence problem Tu & Coley (2022). Although these generative methods facilitate chemical reasoning within a more expansive reaction space, they lack interpretability.

The semi-template-based methods harness the strengths of generative models and pre-existing chemical knowledge. Typical frameworks Yan et al. (2020); Shi et al. (2020); Somnath et al. (2021); Wang et al. (2021); Chen et al. (2023) within this category adhere to a common strategy: they first pinpoint the reaction center and convert the product into synthons. Then, a separate model completes

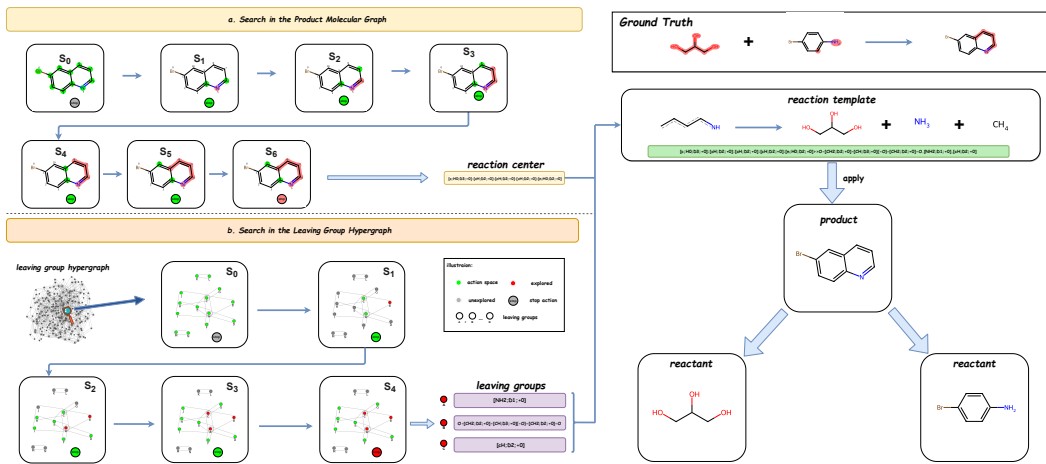

Figure 1: Architecture overview. RretroSiG first identifies reaction center via **(a)** search in the product molecular graph and then completes leaving groups through **(b)** search in the leaving group hypergraph. Finally, RetroSiG converts predicted subgraph into reaction center SMARTS and leaving groups SMARTS, subsequently merging them to derive one retrosynthesis template. RetroSiG obtains the reactants by applying the merged template to the given product. At state $t$, the red highlighted part represents the explored nodes, and the green nodes denote the action space after applying the one-hop constraint.

the transformation of these synthons into reactants. These methods offer competitive accuracy and maintain interpretability due to their stage-wise nature. Nonetheless, they continue to face challenges in predicting reactions with multiple reaction center and fail in the scenario where the same leaving group is attached to more than one atom in a molecular graph. This paper refers to the reaction center comprised of multiple bonds or atoms as multiple raction center. Fig. 1 shows a reaction with multiple reaction center and attaching the same leaving group to more than one atom.

Recently, some methodologies Sacha et al. (2021); Zhong et al. (2023) have framed retrosynthesis prediction as the task of editing the product molecular graph to the reactant molecular graph. In this paper, we refer to this type of method as the graph edit sequence conditional generative model. The edit operations encompass actions such as Delete-bond, Change-bond, Change-atom, Attach-leaving-group, etc. These techniques extract edit sequences from ground-truth reaction data and then model the conditional probability distribution over these sequences given one product molecular graph. Despite its mitigation of some template-based method limitations, the graph edit sequence conditional generative model remains challenged. It cannot handle attaching the same leaving group to more than one atom in a molecular graph and also underutilizes the prior of chemical rules.

Inspired by the limitations mentioned above, we propose, a semi-template-based method, the RetroSiG (**Retro**synthesis via **S**earch **i**n (Hyper) **G**raph) framework (Fig. 1) to conduct single-step retrosynthesis prediction. There are two phases in our RetroSiG, i.e., (a) search in the product molecular graph and (b) search in the leaving group hypergraph. **(a) (Fig. 1a)**: We regard the reaction center identification as the search in the product molecular graph. The critical insight is that the single or multiple reaction center must be a node-induced subgraph of the molecular product graph Coley et al. (2019). Therefore, RetroSiG utilizes a reinforcement learning agent with the graph encoder to explore the appropriate node-induced subgraph in the specific product molecular graph, identifying it as the predicted reaction center. At each step, it considers choosing one node in the molecular product graph and adding it to the explored node-induced subgraph as an action. Since the reaction center graph is connected, we impose a one-hop constraint to reduce the search space and enhance overall performance. **(b) (Fig. 1b)**: We regard the leaving group completion as the search in the leaving group hypergraph. Since the unique leaving group patterns derived from the training dataset can cover 99.7% of the test set Somnath et al. (2021), it is feasible to predict leaving group patterns from the training leaving group corpus. Implicit dependencies exist among various leaving groups. Naturally, we can model these dependencies by creating a leaving group hypergraph. Each node denotes one unique leaving group pattern, while a hyperedge connects the correspond-

ing leaving groups that co-occur within a template (reaction). Therefore, RetroSiG also utilizes a reinforcement learning agent with the hypergraph encoder to explore the appropriate node-induced subgraph in the leaving group hypergraph, selecting it as the resulting leaving groups. At each step, it considers choosing one node in the leaving group hypergraph and adding it to the explored node-induced subgraph as an action. In the evaluation set, we observe that 98.9% of the leaving groups that co-occur within a template are contained within the same hyperedges. Thus, we make full use of the prior and impose a one-hop constraint to reduce the search space. **Finally**, we convert two predicted node-induced subgraphs into reaction center SMARTS and leaving groups SMARTS, subsequently merging them to derive one retrosynthesis template. We obtain the reactants by applying the retrosynthesis template to the given product Coley et al. (2019). Comprehensive experiments demonstrate that RetroSiG achieves a competitive result. Furthermore, we conduct experiments to show the capability of RetroSiG in predicting complex reactions. Ablation experiments verify the effectiveness of individual components, including the one-hop constraint of action space and the leaving group hypergraph. In brief, we highlight our main contributions as follows:

- We propose, a semi-template-based method, the RetroSiG framework to conduct single-step retrosynthesis prediction. It has the capability to predicting reactions with multiple reaction center and attaching the same leaving group to more than one atom.

- The key novelty of RetroSiG is viewing the reaction center identification and leaving group completion as the search in the product molecular graph and the leaving group hypergraph respectively. The hypergraph effectively represents the implicit dependencies among leaving groups, and the search in the graph makes full use of the prior, i.e., one-hop constraint.

- Comprehensive experiments demonstrate that RetroSiG achieves a competitive result. Furthermore, we conduct experiments to show the capability of RetroSiG in predicting complex reactions. Ablation experiments verify the effectiveness of individual components.

## 2 RELATED WORK

We summarize the comparison of existing single-step retrosynthesis in Table 2. Compared with TB, RetroSiG does not need to perform **Subgraph Matching** (NP-Hard) since we can obtain the specific position of the reaction center via search in the product molecular graph. Also, the specific position leads to the unique solution after applying the template, and thus RetroSiG does not need the additional model to **ranking Reactants**. RetroSiG can obtain **Explicit Template** and have the capability of predicting complex reactions with **multiple reaction center (RC)** or **attaching the same leaving group (LG) to more than one atom**.

Table 1: Comparison of different baselines in seven dimensions.

|  | Interpretability | Subgraph Matching | Ranking Reactants | Explicit Template | Multi-RC | Attaching the Same LG to More Than One Atom | Extrapolation Ability |
|---|---|---|---|---|---|---|---|
| TB | ✔ | ✔ | ✔ | ✔ | ✔ | ✔ | ✗ |
| TF | ✗ | ✗ | ✗ | ✗ | ✔ | ✔ | ✔ |
| STB | ✔ | ✗ | ✗ | ✔ | ✗ | ✗ | ✔ |
| Graph2Edits | ✔ | ✗ | ✗ | ✔ | ✔ | ✗ | ✔ |
| RetroSiG | ✔ | ✗ | ✗ | ✔ | ✔ | ✔ | ✔ |

**Graph Edit Sequence Conditional Generative Methods** MEGAN Sacha et al. (2021) and Graph2Edits Zhong et al. (2023) express a chemical reaction as a sequence of graph edits $\boldsymbol{E} = (e_0, \cdots, e_t, \cdots, e_T)$ that transform the product molecular graph into the reactant molecular graph. Here, the edits $e$ encompass various actions such as Delete-bond, Change-bond, Change-atom, Attach-leaving-group, and so forth. The probability of a sequence of graph edits $\boldsymbol{E} = (e_0, \cdots, e_t, \cdots, e_T)$ can be factorized to the probabilities conditional on the product molecule $\boldsymbol{p}$ and $\boldsymbol{E}_{<t} = (e_0, \cdots, e_{t-1})$, i.e.,

$$p(\boldsymbol{E} \mid \boldsymbol{p}) = \prod_{t=1}^{T} p\left(e_t \mid \boldsymbol{p}, \boldsymbol{E}_{<t}\right), T \geq 1 \tag{1}$$

When $T = 0$, $\boldsymbol{E} = (e_0)$ and $p(e_0 \mid \boldsymbol{p}) = 1$ because $e_0$ is a fixed start token. These techniques extract edit sequences from ground-truth reaction data and then model the conditional probability distribution over these edit sequences given one product molecular graph. MEGAN Sacha et al. (2021) was the initial attempt to represent reactions as edits sequences for retrosynthesis prediction. However, this approach faced difficulties in reactant generation due to the atomic-level addition operations. It particularly struggled with reactions involving the attachment of large leaving groups. Graph2Edits Zhong et al. (2023) formulate retrosynthesis as a product-intermediates-reactants reaction reasoning process through a series of graph edits. Importantly, in contrast to the MEGAN model, Graph2Edits substitutes the "add-atom" actions with the attachment of substructures, reducing the number of generation steps and thereby enhancing reactant generation efficiency. These methods acquire knowledge of reaction transformation rules to a certain degree, thus improving their interpretability and generalization capabilities in complex reactions. However, It cannot predict reactions with attaching the same leaving group to more than one atom in a molecular graph and also underutilizes the prior of chemical rules.

**Template-based Methods (TB)** The template is a more efficient and interpretable reaction representation Coley et al. (2019). Following constructing a reaction template library, the methods compare a target molecule with these templates and use the matched template to convert product molecules into reactant molecules. Numerous research efforts have introduced various strategies for prioritizing templates. Coley et al. (2017) employed molecular fingerprint similarity between the target product and the corpus compounds to rank the candidate templates. Segler & Waller (2017) utilized a hybrid neural-symbolic model known as Neuralsym to acquire knowledge for a multi-class classification task related to template selection. Dai et al. (2019) regarded the chemical knowledge in reaction templates as logical rules and acquired the conditional joint probability of these rules and reactants through graph embeddings. Chen & Jung (2021) assessed the appropriate local templates within the predicted reaction centers of a target molecule. Additionally, it considered the non-local impacts of chemical reactions by utilizing global reactivity attention. Yan et al. (2022) composes templates by selecting and annotating molecule subgraphs from training templates. Despite their interpretability, template-based methods have restricted applicability due to their incapability to forecast reactions beyond the template library. Additionally, they are unsuitable for large-scale template sets due to the nature of subgraph matching.

**Template-free Methods (TF)** utilize deep generative models to generate reactant molecules directly. Previous studies Lin et al. (2020); Zheng et al. (2019); Tetko et al. (2020); Seo et al. (2021); Kim et al. (2021); Wan et al. (2022) leverage the Transformer architecture Vaswani et al. (2017) and redefine the problem as a sequence-to-sequence translation task, converting from product to reactants. Tu & Coley (2022) substitutes the original sequence encoder with a graph encoder, ensuring the permutation invariance of SMILES. Although these generative methods facilitate chemical reasoning within a more expansive reaction space, they lack interpretability.

**Semi-template-based Methods (STB)** integrate the benefits of both generative models and supplementary chemical knowledge. Given that chemical reactions typically involve modifying only a small portion of the molecular structure, the majority of existing research approaches have decomposed retrosynthesis into a two-step process: initially identifying the reaction center using a graph neural network to create synthons through molecular editing and then completing the synthons into reactants by either a graph generative model Shi et al. (2020), a Transformer Shi et al. (2020); Wang et al. (2021), or a subgraph selection model Somnath et al. (2021). These methods offer competitive accuracy and maintain interpretability due to their stage-wise nature. Nonetheless, they continue to face challenges in predicting reactions with multiple reaction center. They cannot effectively address scenarios where the same leaving group is attached to more than one atom in a molecular graph.

## 3 METHODOLOGY

### 3.1 SEARCH IN THE PRODUCT MOLECULAR GRAPH

This paper uses RDChiral Coley et al. (2019) to extract the super general reaction center. Hence, regardless of whether the single or multiple reaction center must be a node-induced subgraph of the molecular product graph. A product molecular graph $\mathcal{G}_p = (\mathcal{V}_p, \mathcal{E}_p)$ is represented as a set of $|\mathcal{V}_p|$ nodes (atoms) and a set of $|\mathcal{E}_p|$ edges (bonds). The reaction center graph $\mathcal{G}_{rc} = (\mathcal{V}_{rc}, \mathcal{E}_{rc})$ is a node-induced subgraph of the product molecular graph $\mathcal{G}_p$ such that $\mathcal{V}_{rc} \subseteq \mathcal{V}_p$ and $\mathcal{E}_{rc} =$

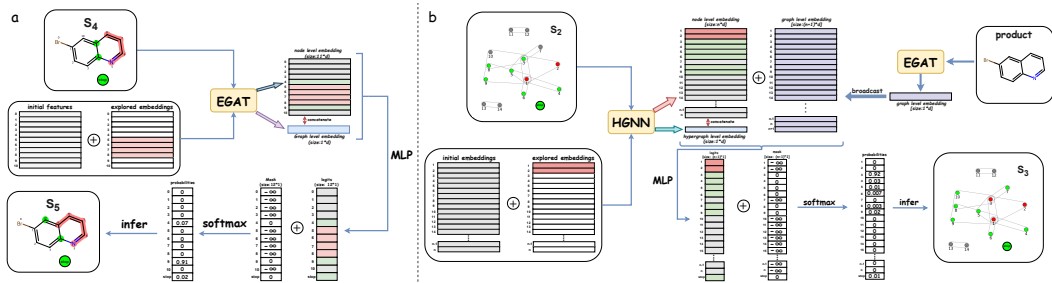

Figure 2: **a.** Policy network architecture in the search in the product molecular graph. **b.** Policy network architecture in the search in the leaving group hypergraph.

$\{(u, v) \mid u, v \in \mathcal{V}_{rc}, (u, v) \in \mathcal{E}_p\}$. Given the product molecular graph $\mathcal{G}_p$, reaction center identification aims to detect the corresponding reaction center graph $\mathcal{G}_{rc}$, i.e. the node set $\mathcal{V}_{rc}$ of $\mathcal{G}_{rc}$. Thus, we regard the reaction center identification as the search in the product molecular graph. RetroSiG considers choosing one node in the molecular product graph and adding it to the explored node-induced subgraph as an action at each step. Ultimately, we transform the identified subgraph into the SMARTS of reaction center.

**State Space:** RetroSiG has a discrete state space $\mathcal{S}$. Each state $s_t \in S$ is represented as $s_t = \{\mathcal{G}_p, \hat{\mathcal{V}}_{rc}^t\}$, where $\hat{\mathcal{V}}_{rc}^t$ denotes the explored node set at step $t$. Notably, $\hat{\mathcal{V}}_{rc}^{t=0} = \emptyset$.

**Action Space:** RetroSiG has two types of actions in its action space $\mathcal{A}$: (1) selecting one node $v$ in the product molecular graph $\mathcal{G}_p$, and (2) stop action, (i.e., doing nothing), denoted as STOP. Since the reaction center graph $\mathcal{G}_{rc}$ is connected, we impose a one-hop constraint on the first type of action. This constraint serves to reduce the search space and enhance overall performance. In other words, the action $a_t$ is to select one node $v$ from the first-order neighbour of the current explored subgraph $\hat{\mathcal{G}}_{rc}^t$ or a STOP action implying stop of the exploration process. At step $t$, action space is represented as $\mathcal{A}^{t>0} = \{v | v \in \text{ONE-HOP}(\hat{\mathcal{G}}_{rc}^t)\} \cup \{\text{STOP}\}$. Notably, $\mathcal{A}^0 = \{v | v \in \mathcal{V}_p\}$.

**Reward Function:** At intermediate step $t$, reward function $\mathcal{R}$ gives $s_t$ a reward 0. At the terminal step $T$, if the predicted $\hat{\mathcal{V}}_{rc}^T$ is identical to the ground-truth $\mathcal{V}_{rc}$, $\mathcal{R}$ gives $s_t$ a reward 1. Otherwise, it gives $s_t$ a reward of 0.

**Policy Net (Fig. 2a):** We use the EGAT Kamiński et al. (2022) to obtain node-level embedding. Each atom $u$ has an initial feature vector $\mathbf{x}_u$. Each bond $(u, v)$ has a feature vector $\mathbf{x}_{uv}$. The details of the initial features can be found in the appendix. For simplicity, we denote the encoding process by $\text{EGAT}(\cdot)$ and describe architectural details in the appendix. The EGAT computes atom representations $\{\mathbf{c}_u^t \mid u \in \mathcal{V}_p\}$ via

$$\{\mathbf{c}_u^t\} = \text{EGAT}\left(\mathcal{G}_p, \left\{\mathbf{x}_u + \mathbb{I}[u \in \hat{\mathcal{V}}_{rc}^t]\mathbf{x}_{explored}\right\}, \{\mathbf{x}_{uv}\}_{v \in \mathcal{N}(u)}\right). \tag{2}$$

In order to incorporate information about the explored node set $\hat{\mathcal{V}}_{rc}^t$ at step $t$, we add a learnable embedding $\mathbf{x}_{explored}$ to the initial feature of the explored node. $\mathbb{I}[u \in \hat{\mathcal{V}}_{rc}^t] \mapsto \{0, 1\}$ be the predicate that indicates whether atom $u$ is the element of $\hat{\mathcal{V}}_{rc}^t$. Here, $\mathcal{N}(u)$ denotes the neighbors of atom $u$. The graph representation $\mathbf{c}_{\mathcal{G}}^t$ is an aggregation of atom representations, i.e. $\mathbf{c}_{\mathcal{G}}^t = \sum_{u \in \mathcal{V}_p} \mathbf{c}_u^t$. Finally, we calculate the logits $s_u^t \in \mathbb{R}^1$, $s_s^t \in \mathbb{R}^1$ for selecting one atom $u$ and STOP action at each step $t$ through the fully connected layers

$$\begin{aligned}
s_u^t &= \boldsymbol{W}_b\left(\sigma\left(\boldsymbol{W}_a\mathbf{c}_u^t + \boldsymbol{B}_a\right)\right) - \mathbb{I}[u \notin \mathcal{A}^t]\infty, \\
s_s^t &= \boldsymbol{W}_b\left(\sigma\left(\boldsymbol{W}_a\mathbf{c}_{\mathcal{G}}^t + \boldsymbol{B}_a\right)\right) - \mathbb{I}[\text{STOP} \notin \mathcal{A}^t]\infty,
\end{aligned} \tag{3}$$

where $\boldsymbol{W}_a$ and $\boldsymbol{W}_b$ are the weights and $\boldsymbol{B}_a$ is the bias. $\mathbb{I}[u \notin \mathcal{A}^t]\,(\mathbb{I}[\text{STOP} \notin \mathcal{A}^t]) \mapsto \{0, 1\}$ be the predicate that indicates whether atom $u$ (STOP) is not in the action space $\mathcal{A}^t$ at step $t$. We add the $-\infty$ to the logit of the action not appearing in the action space $\mathcal{A}^t$ at step $t$. It ensures that the probability of an action not appearing in the action space equals 0 after SOFTMAX operation.

## 3.2 Search in the Leaving Group Hypergrah

In this paper, we use RDChiral Coley et al. (2019) to extract the super general leaving groups from the training set and obtain unique leaving group patterns. Since the unique leaving group patterns derived from the training dataset can cover 99.7% of the test set Somnath et al. (2021), it is feasible to predict leaving group patterns from the training leaving group corpus. Implicit dependencies exist among various leaving groups. Naturally, we build a hypergraph to model these dependencies. Each node denotes one unique leaving group pattern, while a hyperedge connects the corresponding leaving groups that co-occur within a template. A leaving group hypergraph $\mathcal{G}_{hg} = (\mathcal{V}_{hg}, \mathcal{E}_{hg})$ is represented as a set of $|\mathcal{V}_{hg}|$ nodes (unique leaving group) and a set of $|\mathcal{E}_{hg}|$ hyperedges (co-occurrence). Given the product molecular graph $\mathcal{G}_p$, leaving group completion aims to identify the corresponding leaving groups, i.e. the sub-node set $\mathcal{V}_{lg}$ of $\mathcal{V}_{hg}$. Thus, we regard the leaving group completion as the search in the leaving group hypergraph. RetroSiG considers choosing one node in the leaving group hypergraph and adding it to the explored node set as an action at each step. In the end, we transform the identified nodes into the SMARTS of leaving groups.

**State Space:** Similar to search in the product molecular graph, each state $s_t \in S$ is represented as $s_t = \{\mathcal{G}_p, \mathcal{G}_{hg}, \hat{\mathcal{V}}_{lg}^t\}$, where $\hat{\mathcal{V}}_{lg}^t$ denotes the explored node set at step $t$. Notably, $\hat{\mathcal{V}}_{lg}^{t=0} = \emptyset$.

**Action Space:** Here, RetroSiG also has two types of actions in its action space $\mathcal{A}$: (1) selecting one node $v$ in the leaving group hypergraph $\mathcal{G}_{hg}$, and (2) stop action, (i.e., doing nothing), denoted as STOP. In the evaluation set, 98.9% of the leaving groups that co-occur within a template are contained within the $\mathcal{E}_{hg}$. It implies that, during inference, the prediction is likely to be incorrect if the chosen leaving group originates from different hyperedges. Thus, we impose a one-hop constraint on the first type of action. This constraint serves to reduce the search space and enhance overall performance. In other words, the action $a_t$ is to select one node $v$ from the first-order neighbour of the current explored node set $\hat{\mathcal{V}}_{lg}^t$ or a STOP action. At step $t$, action space is represented as $\mathcal{A}^{t>0} = \{v | v \in \text{ONE-HOP}(\hat{\mathcal{V}}_{lg}^t\} \cup \{\text{STOP}\}$. Notably, $\mathcal{A}^0 = \{v | v \in \mathcal{V}_{hg}\}$.

**Reward Function:** At intermediate step $t$, reward function $\mathcal{R}$ gives $s_t$ a reward 0. At the terminal step $T$, if the predicted $\hat{\mathcal{V}}_{lg}^T$ is identical to the ground-truth $\mathcal{V}_{lg}$, $\mathcal{R}$ gives $s_t$ a reward 1. Otherwise, it gives $s_t$ a reward of 0.

**Policy Net (Fig. 2b):** We use the HGNN Feng et al. (2019) to obtain node-level embedding. We assign each atom $u$ in the hypergraph a randomly initialized learnable embedding $\mathbf{f}_u$. For simplicity, we denote the encoding process by $\text{HGNN}(\cdot)$ and describe architectural details in the appendix. The HGNN computes atom representations $\{\mathbf{h}_u^t \mid u \in \mathcal{V}_{hg}\}$ via

$$\{\mathbf{h}_u^t\} = \text{HGNN}\left(\mathcal{G}_{hg}, \left\{\mathbf{f}_u + \mathbb{I}[u \in \hat{\mathcal{V}}_{lg}^t]\mathbf{f}_{explored}\right\}\right). \tag{4}$$

In order to incorporate information about the explored node set $\hat{\mathcal{V}}_{lg}^t$ at step $t$, we add a learnable embedding $\mathbf{f}_{explored}$ to the initial embedding of the explored node. $\mathbb{I}[u \in \hat{\mathcal{V}}_{lg}^t] \mapsto \{0, 1\}$ be the predicate that indicates whether atom $u$ is the element of $\hat{\mathcal{V}}_{lg}^t$. The graph representation $\mathbf{h}_{\mathcal{G}}^t$ is an aggregation of atom representations, i.e. $\mathbf{h}_{\mathcal{G}}^t = \sum_{u \in \mathcal{V}_{hg}} \mathbf{h}_u^t$. We also use the EGAT Kamiński et al. (2022) to obtain graph-level embedding $\mathbf{c}_{\mathcal{G}}$ of the product molecular graph $\mathcal{G}_p$. Finally, we calculate the logits $s_u^t \in \mathbb{R}^1$, $s_s^t \in \mathbb{R}^1$ for selecting one atom $u$ and STOP action at each step $t$ through the fully connected layers

$$\begin{aligned} s_u^t &= \mathbf{W}_b \left(\sigma \left(\mathbf{W}_a \left(\mathbf{h}_u^t + \mathbf{c}_{\mathcal{G}}\right) + \mathbf{B}_a\right)\right) - \mathbb{I}[u \notin \mathcal{A}^t]\infty, \\ s_s^t &= \mathbf{W}_b \left(\sigma \left(\mathbf{W}_a \left(\mathbf{h}_{\mathcal{G}}^t + \mathbf{c}_{\mathcal{G}}\right) + \mathbf{B}_a\right)\right) - \mathbb{I}[\text{STOP} \notin \mathcal{A}^t]\infty, \end{aligned} \tag{5}$$

where $\mathbb{I}[u \notin \mathcal{A}^t]$ ($\mathbb{I}[\text{STOP} \notin \mathcal{A}^t]) \mapsto \{0, 1\}$ be the predicate that indicates whether atom $u$ (STOP) is not in the action space $\mathcal{A}^t$ at step $t$. The $-\infty$ operation ensures that the probability of an action not appearing in the action space equals 0 after SOFTMAX operation.

## 3.3 Training and Inference

The policy network learns to maximize the expected reward and improves search quality over episodes, which is updated by Proximal Policy Optimization (PPO) Schulman et al. (2017):

$L^{CLIP}(\theta) = \hat{E}_t[\min(r_t(\theta)\hat{A}_t, \text{clip}(r_t(\theta), 1 - \epsilon, 1 + \epsilon)\hat{A}_t)]$. Here, $\hat{E}_t$ represents the expected value at timestep t, $r_t(\theta)$ is the ratio of the new policy and the old policy, and $\hat{A}_t$ is the estimated advantage at timestep t.

Inference is performed using beam search Ma et al. (2021) with a log-likelihood scoring function. For a beam size $k$, we can obtain two sets of $k$ candidate solutions from two stages. Finally, we have $k^2$ candidate solutions. From the $k^2$ possibilities, we select $k$ candidate solutions with the highest cumulative log-likelihood score as the top-k prediction results.

## 4 EVALUATION

### 4.1 DATA

We employ the traditional retrosynthesis benchmark dataset USPTO-50K Schneider et al. (2016) to evaluate our approach. It contains 50016 reactions with the correct atom-mapping, classified into ten distinct reaction types. We follow the identical data split described in Coley et al. (2017), dividing it into 40k, 5k, and 5k reactions for the training, validation, and test sets. To eliminate the information leakage in the USPTO-50K dataset, as previously noted in related studies Somnath et al. (2021); Yan et al. (2020), we additionally canonicalize the product SMILES and re-assign the atom-mapping to the reactant atoms following the method given by Somnath et al. (2021).

### 4.2 EVALUATION METRIC

Consistent with prior research, we employ top-k exact match accuracy as our evaluation metrics. We used k with 1, 3, 5, and 10 for comparison purposes in our experiments. This metric compares whether the predicted set of reactants are exactly the same as ground truth reactants. Implementation details can be found in the appendix. The source code can be found in the supplementary materials.

### 4.3 BASELINE

We compare the prediction results of RetroSiG with several template-based, template-free, and semi-template-based methods. Notably, we introduce semi-template-based methods including G2G Shi et al. (2020), RetroXpert Yan et al. (2020), RetroPrime Wang et al. (2021), MEGAN Sacha et al. (2021), GraphRetro Somnath et al. (2021), and Graph2Edits Zhong et al. (2023) as primary baselines due to their exceptional performance. We also considered template-based models like Retrosim Coley et al. (2017), Neuralsym Segler & Waller (2017), GLN Dai et al. (2019), LocalRetro Chen & Jung (2021), and template-free models such as SCROP Zheng et al. (2019), Augmented Transformer Tetko et al. (2020), GTA Seo et al. (2021), Graph2SMILES Tu & Coley (2022), Dual-TF Sun et al. (2021), Retroformer Wan et al. (2022) as robust baseline models for comparison.

### 4.4 PERFORMANCE

**Main Results:** The top-k exact match accuracy results on the USPTO-50k benchmark are shown in Table 2. When the reaction class is unknown, our method achieves a 54.9% top-1 accuracy which is competitive and only 0.2% lower than the SOTA (Graph2Edits) result of 55.1%. However, for k equals 3 and 5, RetroSiG outperforms the performance of Graph2Edits. RetroSiG also beats LocalRetro by a margin of 1.5% and 0.1% respectively in top-1 and top-3 accuracy. With the reaction class given, our method achieves a 66.5% top-1 accuracy which is only 0.6% lower than the SOTA (Graph2Edits). For k equals 3, 5 and 10, RetroSiG outperforms the performance of Graph2Edits. We attribute it to the solid prior knowledge carried by the one-hop mechanism.

**Complex Sample Results:** We investigate the performance effect of some complex reactions in the USPTO-50k, including reactions with multiple reaction center and attaching the same leaving group to more than one atom. As is shown in Fig. 3a, Most reactions with 1, 2 atoms in the reaction center account for the majority, which have 1121 (22.3%) and 3604 (72.0%) cases respectively. However, the reactions with 3, 4, 5, 6, and $\geq 7$ account for a small proportion, which have 184 (3.7%), 29 (0.6%), 24 (0.5%), 36 (0.7%) and 9 (0.2%) cases respectively. Multiple reaction center consists of three or more atoms, and 92.6% (261 / 282) of the samples with multiple reaction center are with attaching the same leaving group to more than one atom. Thus, we report the top-10 accuracy of the

Table 2: Top-$k$ accuracy for retrosynthesis prediction on USPTO-50K.

| Model | Top-k accuracy (%) | | | | | | | |
|---|---|---|---|---|---|---|---|---|
| | Reaction class unknown | | | | Reaction class known | | | |
| | k=1 | 3 | 5 | 10 | 1 | 3 | 5 | 10 |
| Template-Based Methods | | | | | | | | |
| Retrosim | 37.3 | 54.7 | 63.3 | 74.1 | 52.9 | 73.8 | 81.2 | 88.1 |
| Neuralsym | 44.4 | 65.3 | 72.4 | 78.9 | 55.3 | 76.0 | 81.4 | 85.1 |
| GLN | 52.5 | 69.0 | 75.6 | 83.7 | 64.2 | 79.1 | 85.2 | 90.0 |
| LocalRetro | 53.4 | 77.5 | 85.9 | 92.4 | 63.9 | 86.8 | 92.4 | 96.3 |
| Template-Free Methods | | | | | | | | |
| SCROP | 43.7 | 60.0 | 65.2 | 68.7 | 59.0 | 74.8 | 78.1 | 81.1 |
| Aug.Transformer | 53.2 | - | 80.5 | 85.2 | - | - | - | - |
| GTA | 51.1 | 67.6 | 74.8 | 81.6 | - | - | - | - |
| Graph2SMILES | 52.9 | 66.5 | 70.0 | 72.9 | - | - | - | - |
| Dual-TF | 53.6 | 70.7 | 74.6 | 77.0 | 65.7 | 81.9 | 84.7 | 85.9 |
| Semi-Template-Based Methods | | | | | | | | |
| G2G | 48.9 | 67.6 | 72.5 | 75.5 | 61.0 | 81.3 | 86.0 | 88.7 |
| RetroXpert | 50.4 | 61.1 | 62.3 | 63.4 | 62.1 | 75.8 | 78.5 | 80.9 |
| RetroPrime | 51.4 | 70.8 | 74.0 | 76.1 | 64.8 | 81.6 | 85.0 | 86.9 |
| MEGAN | 48.1 | 70.7 | 78.4 | 86.1 | 60.7 | 82.0 | 87.5 | 91.6 |
| GraphRetro | 53.7 | 68.3 | 72.2 | 75.5 | 63.9 | 81.5 | 85.2 | 88.1 |
| Graph2Edits | 55.1 | 77.3 | 83.4 | 89.4 | 67.1 | 87.5 | 91.5 | 93.8 |
| RetroSiG | 54.9 | 77.6 | 84.1 | 89.0 | 66.5 | 87.9 | 92.0 | 94.1 |

number of atoms 3, 4, 5, 6, and $\geq 7$ in Fig. 3b to investigate the performance of complex reactions. We can see that the performance of our model does not decrease significantly with the increasing number of atoms. It demonstrates that our model can handle these complex reactions.

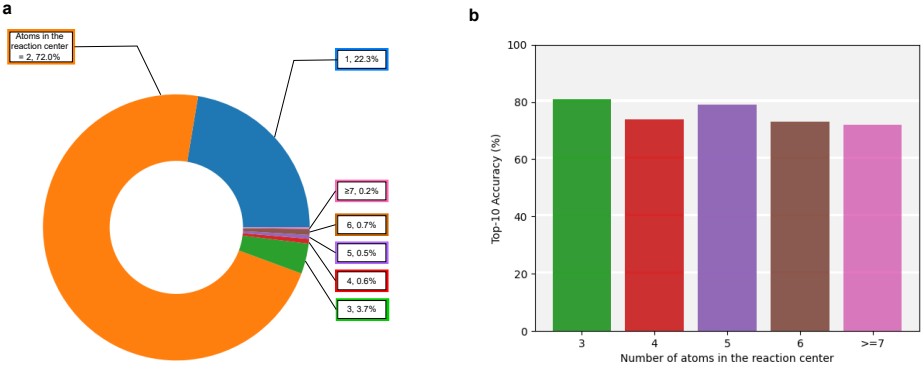

Figure 3: Analysis of complex samples.

**Ablation Study:** In Fig. 3, Hypergraph-Structure ✔ denotes that we use HGNN and hypergraph structure to encode each node (leaving group) embedding. One-Hop Constraint ✔ denotes that we use a one-hop constraint. ① outperforms ② and ③, showing that one-hop constraint is effective. ② outperforms ③, demonstrating that one-hop constraint in reaction center identification contributes more to performance improvement. ④ outperforms ⑤, implying that hypergraph effectively models dependencies between leaving groups.

Table 3: Ablation Study.

| | Hypergraph-Structure | One-Hop Constraint(rc) | One-Hop Constraint(lg) | Top-1(%) |
|---|---|---|---|---|
| ① | ✔ | ✔ | ✔ | 54.9 |
| ② | ✔ | ✔ | ✘ | 54.6 |
| ③ | ✔ | ✘ | ✔ | 53.5 |
| ④ | ✔ | ✘ | ✘ | 53.4 |
| ⑤ | ✘ | ✘ | ✘ | 52.8 |

## 4.5 VISUALIZATION OF EXAMPLE PREDICTIONS

In Fig. 4, we visualize the model predictions and the ground truth for three cases. Fig. 4a shows an example with attaching the same leaving group to more than one atom. Moreover, the model identifies both the reaction center and leaving groups correctly. Fig. 4b, the model identifies the reaction center correctly, while the predicted leaving groups are incorrect. We hypothesize this is because carboxyl (COOH) is more common in the training set than acyl chloride (COCl). However, the prediction results of top-1 are also feasible chemically. Fig. 4c, the model incorrectly identifies the reaction center and consequently the leaving group, resulting in top-1 prediction being inconsistent with ground truth. Top-1 prediction is an open-loop operation, which is not necessarily harmful from the perspective of multi-step retrosynthesis. Interestingly, the reaction of top-1 prediction is to attach more than one leaving group to one synthon. Previous semi-template-base methods Somnath et al. (2021) could not predict such complex samples.

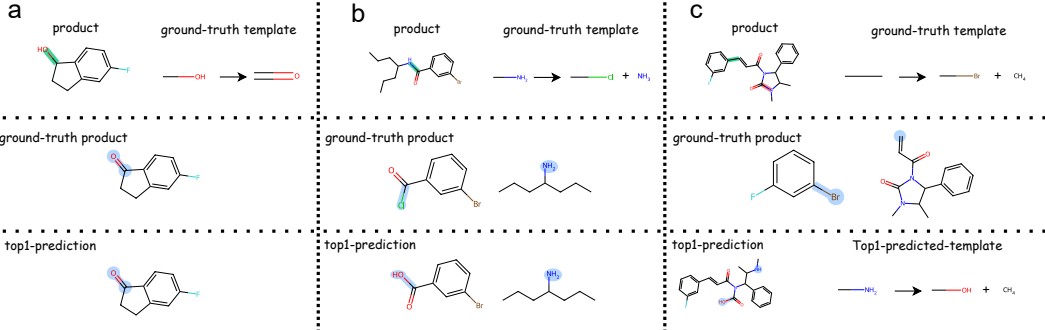

Figure 4: Visualization of Example Predictions.

## 5 DISCUSSION AND CONCLUSION

**Limitations:** We share one limitation of all semi-template-based and graph-edits methods. They are highly reliant on atom-mapping information between products and reactants. Incorrect mapping will extract the wrong template. It will provide incorrect reaction center labels and leaving group labels during model training. We look forward to the continuous improvement of the atom-mapping algorithm and the extraction/application template algorithm of Rdchiral Coley et al. (2019) in the community, which is very beneficial to the RetroSiG framework.

**Future Works:** There are three directions for future work. **End2End:** We need to design an end-to-end architecture to connect the two search processes so that the signal at the final step can update both stages. **Decision Transformer :** Our current policy net consists of a simple (hyper) graph encoder and MLP; hence, we must design a more powerful policy net. The ability of Transformer Vaswani et al. (2017) has recently been proven in various fields, especially in large language models Ouyang et al. (2022). Naturally, Decision Transformer Chen et al. (2021) is a choice as a policy net. **Termination Reward:** At termination time, the environment rewards the agent only when the prediction is exactly the same as the ground truth. However, multiple valid ways can synthesize a product. We need to design a more reasonable termination reward.

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

## A  ATOM AND BOND FEATURES

In this paper, initial features of atom and bond can be found in Table 4 and Table 5.

Table 4: Atom Features used in EGAT. All features are one-hot encoding.

| Feature | Description | Size |
|---------|-------------|------|
| Atom type | Type of an atom by atomic number. | 100 |
| Total degree | Degree of an atom including Hs. | 6 |
| Explicit valence | Explicit valence of an atom. | 6 |
| Implicit valence | Explicit valence of an atom. | 6 |
| Hybridization | sp, sp2, sp3, sp3d, or sp3d2. | 5 |
| # Hs | Number of bonded Hydrogen atom. | 5 |
| Formal charge | Integer electronic charge assigned to atom. | 5 |
| Aromaticity | Whether an atom is part of an aromatic system. | 1 |
| In ring | Whether an atom is in ring | 1 |

Table 5: Bond features used in EGAT. All features are one-hot encoding.

| Feature | Description | Size |
|---------|-------------|------|
| Bond type | Single, double, triple, or aromatic. | 4 |
| Conjugation | Whether the bond is conjugated. | 1 |
| In ring | Whether the bond is part of a ring. | 1 |
| Stereo | None, any, E/Z or cis/trans. | 6 |
| Direction | The direction of the bond. | 3 |

## B  EDGE GRAPH ATTENTION NETWORK LAYER

Given $\mathbf{H}^{(t)} = [\boldsymbol{h}_1^{(t)}; \boldsymbol{h}_2^{(t)}; \cdots ; \boldsymbol{h}_n^{(t)}] \in R^{n \times d}$ and $\mathbf{f}_{ij}^{(t)} \in R^{1 \times d}$ are the node embedding matrix and the embedding of edge $(i, j)$ at the $t$-th layer repectively, where $\boldsymbol{h}_i^{(t)} \in R^{1 \times d}$ is the node-level embedding for node $i$ of the graph and is also the $i$-th row of $\mathbf{H}^{(t)}$, $d$ is the dimension of node-level embedding and $n$ is the number of nodes. $\boldsymbol{h}_i^{(0)}$ and $\boldsymbol{f}_{ij}^{(0)}$ are the initial features of node and edge in molecular product graph $\mathcal{G}_p$. EGAT injects the graph structure into the attention mechanism by performing masked attention, namely it only computes $\alpha_{ij}$ for nodes $j \in \mathcal{N}_i$, where $\mathcal{N}_i$ is the first-order neighbors of node $i$ in the graph:

$$\boldsymbol{f}_{ij}^{(t+1)} = \text{LeakyReLU}\left(\left[\boldsymbol{h}_i^{(t)}\mathbf{W} \left\| \boldsymbol{f}_{ij}^{(t)} \right\| \boldsymbol{h}_j^{(t)}\mathbf{W}\right] A\right),$$
$$e_{ij} = \mathbf{a} \cdot \boldsymbol{f}_{ij}^{(t+1)^T}, \alpha_{ij} = \frac{\exp(e_{ij})}{\sum_{k \in \mathcal{N}_i} \exp(e_{ik})}, \tag{6}$$

where $e_{ij} \in R$ and $\alpha_{ij} \in R$ are a non-normalized attention coefficient and a normalized attention coefficient representing the weight of message aggregated from node $j$ to node $i$ respectively in the $t$-th layer of EGAT, and $\|$ is the concatenation operation. Besides, $\mathbf{W} \in R^{d \times d}$, $A \in R^{3d \times d}$ and $\mathbf{a} \in R^{1 \times d}$ are learnable parameters in the $t$-th layer.

EGAT employs multi-head attention to stabilize the learning process of self-attention, similar to Transformer Vaswani et al. (2017). If there are $K$ heads, $K$ independent attention mechanisms

execute the Eq. 6, and then their features are concatenated:

$$\boldsymbol{h}_i^{(t+1)} = \text{MLP}\left(\|_{k=1}^{K}\sigma\left(\sum_{j\in\mathcal{N}_i}\alpha_{ij}^k\boldsymbol{h}_j^{(t)}\mathbf{W}^k\right)\right) \tag{7}$$

where $\|$ represents concatenation, $\alpha_{ij}^k$ are normalized attention coefficients computed by the $k$-th learnable $\mathbf{W}^k \in R^{d\times d}$, $A^k \in R^{3d\times d}$ and $\mathbf{a}^k \in R^{1\times d}$ following Eq. 6. Besides, MLP denotes multi-perceptron.

## C  HYPER GRAPH NEURAL NETWORK LAYER

Specifically, for the $l$-th layer in HGNN, it takes hypergraph $\mathcal{G}_{hg}$'s incidence matrix H and hidden representation matrix $\mathrm{X}^l$ as input, then the node representations in next layer will be computed as follows:

$$\mathbf{X}^{l+1} = \text{HGNN}\left(\mathbf{X}^l, \mathbf{H}, \mathbf{\Theta}^l\right) = \sigma\left(\mathbf{D}_v^{-\frac{1}{2}}\mathbf{H}\mathbf{W}\mathbf{D}_e^{-1}\mathbf{H}^\top\mathbf{D}_v^{-\frac{1}{2}}\mathbf{X}^l\mathbf{\Theta}^l\right) \tag{8}$$

where $\sigma(\cdot)$ is the nonlinear activation function. $\mathbf{D}_v, \mathbf{D}_e, \mathbf{W}$ are the diagonal node degree, edge degree and edge weight matrices, respectively. $\mathbf{\Theta}^l$ is a trainable parameter matrix.

## D  IMPLEMENTATION DETAILS

Our proposed RetroSiG is implemented with Deep Graph Library (DGL) Wang et al. (2019), Deep-Hypergraph[1] (DHG) and Pytorch Paszke et al. (2019). As for the EGAT, we stack four identical four-head attentive layers of which the hidden dimension is 512. As for the HGNN, we stack two layers of which the hidden dimension is 512. All embedding sizes in the model are set to 512. We use $\text{ELU}(x) = \alpha(\exp(x) - 1)$ for $x \leq 0$ and $x$ for $x > 0$ as our activation function where $\alpha = 1$ as our activation function. We conduct all the experiments on a machine with an Intel Xeon 4114 CPU and one Nvidia Titan GPU. For training, we set the learning rate to 0.0001, the number of training iterations to 10,000,000, and use the Adam optimizer Kingma & Ba (2014). Checkpoints are saved for each 1000 iteration to select the best checkpoints on the evaluation set. The source code can be found in the supplementary materials.

---

[1]https://github.com/iMoonLab/DeepHypergraph

