# OpenReview forum: "Retrosynthesis Prediction via Search in (Hyper) Graph"
_ICLR.cc/2024/Conference — ICLR 2024 Conference Withdrawn Submission_

### Official Review · Reviewer_nw3M · 2023-10-21

**Soundness:** 2 fair
**Presentation:** 2 fair
**Contribution:** 3 good
**Rating:** 5
**Confidence:** 5

**Summary:**

This paper models the retrosynthesis prediction as a search problem. In the first stage, the method searches the reaction center in the product molecule graph. In the second stage, the method searches the leaving groups in the hypergraph where leaving groups within the same reaction template are connected with the hyperedges. In the third stage, the method merges the predicted reaction center and leaving groups to obtain a predicted template. After that, the method applies the predicted template on the product to get the predicted reactants.

**Strengths:**

1. The writing is clear.

2. The method is novel to me. The method transforms the template prediction into a search problem.

3. The method can handle the reaction with multiple reaction centers, which is ignored by the previous semi-templated-based methods.

**Weaknesses:**

1. First of all, I do think the illustration figures are too small for me to understand their meaning.

2. In essence, this paper introduces a method for template prediction. However, it still faces challenges with generalization. If a template from the test set isn't present in the training set, its associated leaving groups won't be linked by hyperedges within the hypergraph. Consequently, when employing a search limited by a one-hop constraint, searching the relevant leaving groups becomes challenging.

3. While extracting leaving groups from the training set poses generalization challenges, a more effective method involves constructing leaving groups directly from the molecules.

4. Using reinforcement learning might be superfluous. I suggest modeling the search for the reaction center and leaving groups as an autoregressive problem.

5. Evaluating performance on the USPTO-50K dataset, which mostly contains reactions with a single reaction center, doesn't convincingly show that the introduced method can handle cases with multiple reaction centers. A recent study by FusionRetro[1] proposed a dataset from USPTO-full for retrosynthetic planning, where approximately 30% of the reactions have several reaction centers. Both G2Gs[2] and GraphRetro[3] demonstrate subpar performance on this dataset, primarily because they struggle with multi-reaction-center scenarios. I want to know the performance of RetroSiG on this dataset to validate its effectiveness in handling multi-reaction-center reactions.

[1] FusionRetro: Molecule Representation Fusion via In-Context Learning for Retrosynthetic Planning, ICML 2023.
[2] A graph to graphs framework for retrosynthesis prediction, ICML 2020.
[3] Learning graph models for retrosynthesis prediction. NeurIPS 2021.

**Questions:**

N/A

---

### Official Review · Reviewer_tzVj · 2023-11-01

**Soundness:** 3 good
**Presentation:** 3 good
**Contribution:** 2 fair
**Rating:** 5
**Confidence:** 4

**Summary:**

This paper introduces RetroSiG, a new method for predicting reactants in organic synthesis, specifically designed for complex reactions with multiple reaction centers or multiple attachments of the same leaving group. RetroSiG utilizes a semi-template-based approach, employing a search mechanism in the product molecular graph and a leaving group hypergraph to identify reaction centers and complete leaving groups. The method leverages the hypergraph to capture implicit dependencies between leaving groups and incorporates a one-hop constraint to reduce the search space and improve performance. The paper includes comprehensive experiments demonstrating RetroSiG's competitive performance in predicting complex reactions. Ablation experiments validate the effectiveness of individual components, such as the one-hop constraint and the leaving group hypergraph. Overall, RetroSiG overcomes the limitations of existing methods and offers advantages in handling complex reactions, modeling dependencies, and utilizing prior knowledge.

**Strengths:**

S1. The paper demonstrates a level of originality in addressing the challenge of retrosynthesis prediction in organic synthesis. While semi template-based and graph-edits-based methods have been previously explored, the paper introduces a novel approach called RetroSiG, which combines a semi-template-based method with a search mechanism in the product molecular graph and leaving group hypergraph. This integration aims to overcome the limitations of existing methods in predicting complex reactions with multiple reaction centers or the attachment of the same leaving group to multiple atoms. The combination of these elements appears to be a novel contribution to the field.

S2. RetroSiG highlights the use of a semi-template-based approach, the search mechanisms in the product molecular graph and leaving group hypergraph, and the incorporation of a one-hop constraint.

S3. The significance of the paper lies in its attempt to address the limitations of existing methods in predicting complex reactions in retrosynthesis. By proposing RetroSiG, which combines various techniques and incorporates a one-hop constraint, the paper aims to enhance the accuracy and interpretability of retrosynthesis prediction. Experiments validated the claims, that RetroSiG has practical implications in the field of organic synthesis and contributes to the development of more effective methods for predicting reactants.

**Weaknesses:**

W1. Incorporate more diverse and complex reaction datasets: You could improve the generalizability of your method by incorporating more diverse and complex reaction datasets. This would help to ensure that the method can handle a wider range of reactions and produce more accurate predictions.

W2. Provide a more detailed analysis of the experimental results: While you provide some experimental results, you could provide a more detailed analysis of the results to help readers better understand the performance of the proposed method. For example, you could analyze the performance of the method on different types of reactions or provide a more detailed analysis of the errors made by the method.

**Questions:**

Q1. How does the proposed method handle cases where the reactants are not unique or where there are multiple possible reactants that can lead to the same product? Can you provide some examples of such cases and how the proposed method performs in these cases?

Q2. The data set and hypergraph search method used in this article are aimed at single-step retrosynthesis and have produced good results. Can this method continue to be extended to multi-step retrosynthesis?

Q3. Does this method give good predictions for reactions involving chiral molecules (one of the chemical bond characteristics in the appendix)?

---

### Official Review · Reviewer_2mmA · 2023-11-01

**Soundness:** 3 good
**Presentation:** 1 poor
**Contribution:** 2 fair
**Rating:** 5
**Confidence:** 4

**Summary:**

This paper proposes a semi-template-based model for (backward) retrosynthesis prediction. This method is based on predicting the reaction centre using an RL agent selecting a connected subgraph atom-by-atom, followed by another agent selecting a subset of leaving groups, which is again constrained to be connected in a special co-occurrence hypergraph. The authors then experiment on USPTO-50K to showcase the effectiveness of their approach.

**Strengths:**

(S1): The retrosynthesis prediction problem is generally important, and the authors correctly note how some of the existing models can be too constrained with respect to either the form of the reaction centre or multiset of leaving groups.

(S2): The high-level approach is reasonable (although I have many comments about specific details – see (W1)).

**Weaknesses:**

(W1): Several parts of the method are either not clear to me, or they are clear but I am not sure if they are necessary.

- (a) How does RetroSiG choose where the leaving groups are attached in the graph, and also how many times each group is used? The action space used by the hypergraph agent suggests each node (leaving group) is only chosen once, so the count of how many it is applied has to be selected separately.

- (b) The authors mention that the final prediction is produced by assembling a template and applying it, but that this is more efficient than template application in template-based models as the location where the template should be applied is known. How is the template application carried out technically? Initially I assumed it's through a standard library e.g. rdkit or rchiral, but normally those would try to match the template in all places. How do you restrict the matching to only consider the predicted reaction centre?

- (c) Are the only rewards received by the RL agents coming from choosing the right set of atoms/nodes at the end? If so, why is RL needed at all? It seems one could just sample a possible correct sequence of actions (or several such sequences) and supervise the model directly, as done in e.g. MEGAN or models for molecular graph generation [1].

- (d) Is using a hypergraph for the leaving groups necessary, as opposed to just using a plain graph where two leaving groups are connected if they co-occur (as a pair) in the training set? As far as I understand, the allowed actions would be the same, as such a graph would have the same connectivity as the hypergraph; however, the encoder would be different (GNN vs Hypergraph GNN), so maybe it would make some difference in performance.

- (e) How is reaction centre defined? It may sometimes be the case that the set of bond changes is not a connected subgraph, hence the atoms with changing neighborhood may also not form a connected set of vertices. In those cases, standard template extraction algorithms might extract a template with several "connected components". However, as I understand RetroSiG assumes the set of nodes in the reaction centre is connected, so perhaps the definition is different than "nodes for which neighborhood changed"?

- (f) Is the second step (leaving group identification) not conditioned on the choices made in the first step in any way?

(W2): Empirical performance as shown in Table 2 is promising, but some further comparison would help to really determine where the proposed method shines.

- (a) I like how the authors isolate a "complex subset" where the reaction centre has more than 2 atoms, even though on USPTO-50K this unfortunately leaves very few samples. However, the authors only run RetroSiG on that subset, and do not run the other baseline models. I think it would be informative to compare with other models and see if e.g. RetroSiG starts outperforming some models it was losing to on the full dataset.

- (b) Further from the above, the tiny size of the "complex subset" in USPTO-50K again points at the fact that this dataset is relatively simple, and thus a method designed to e.g. better deal with larger reaction centres or more complex leaving group patterns may not show its full potential. It would therefore be good to e.g. run comparison on a larger version of USPTO like USPTO-MIT. One could even repeat the same "complex subset" selection on the larger dataset for another comparison. This would be helpful to understand if RetroSiG really is more adept at dealing with more complex reactions.

- (c) Some SotA approaches are missing: RetroKNN [2] on the template-based side, and RootAligned [3] on the template-free side.


=== Other comments ===

(O1): There are several caveats that can distort accuracies in reaction prediction; the authors note one of them (careful atom mapping removal to avoid data leakage) but there are other subtleties (e.g. output deduplication) which could be taken into account. See [4] for best practices on this; it would be good to verify that RetroSiG followed these, as [4] shows prior works often did not and that distorted some of the numbers.

(O2): Section 4.5 points to Figure 4a for an example of attaching the same leaving group to more than one atom, yet in the figure I can see the group only attached once. Am I misunderstanding something here or is there an error in the figure?

(O3): Next to Equation 1 there is some text about what happens when T = 0. Why does one have to consider such a case? Does this mean the sequence of edits is an empty sequence i.e. product and reactants are the same?



=== Nitpicks ===

Below I list nitpicks (e.g. typos, grammar errors), which did not have a significant impact on my review score, but it would be good to fix those to improve the paper further.

- Most citations seem to lack parentheses; this form of citation should only be used for citations that are part of text (e.g. "Anon et al proposed..." but not for citations that are outside of text "Some models (Anon et al) compute..."). Please check the use of `\cite` vs `\citet` vs `\citep` commands.

- "we propose," (two places) -> I would not put a comma here

- "RetroSiG has several advantages as a semi-template-based method:  First" -> I would just put a period instead of a colon here

- "RretroSiG"

- "raction center"

- "It has the capability to predicting" -> "to predict"

- "can be factorized to" -> "into"

- "However, It cannot predict" -> lowercase "it"

- "from the first-order neighbour" -> "neighbours"

- "As is shown in Fig. 3a, Most" -> lowercase "most"

- The ICLR LaTeX guidelines are to remove most horizontal lines from tables, so I would do that in Table 2. Maybe leave only those separating the different groups.

- Ablation study points to Figure 3, but I think it should point to Table 3

- Figure 4 says "ground-truth product" but I think it should say "ground-truth reactants"



=== References ===

[1] "Learning to Extend Molecular Scaffolds with Structural Motifs"

[2] "Retrosynthesis Prediction with Local Template Retrieval"

[3] "Root-aligned SMILES: A Tight Representation for Chemical Reaction Prediction"

[4] "Re-evaluating Retrosynthesis Algorithms with Syntheseus"

**Questions:**

See the "Weaknesses" section above for specific questions.